# Factors Influencing Sleep Quality in Open-Heart Patients in the Postoperative Intensive Care Unit

**DOI:** 10.3390/healthcare10112311

**Published:** 2022-11-18

**Authors:** Ting-Ru Lin, Ching-Hui Cheng, Jeng Wei, Tsae-Jyy Wang

**Affiliations:** 1Department of Nursing, Cardinal Tien College of Healthcare and Management, New Taipei City 231, Taiwan; 2Cheng Hsin General Hospital, Taipei City 112, Taiwan; 3School of Nursing, National Taipei University of Nursing and Health Sciences, Taipei City 112, Taiwan

**Keywords:** sleep quality, open-heart patients, intensive care unit

## Abstract

Open-heart patients often experience sleep problems postoperatively. This cross-sectional study is aimed to investigate open-heart patients’ sleep quality and its influencing factors during intensive care. A consecutive sample of 117 eligible open-heart patients was recruited from an intensive care unit (ICU) of a general hospital. Data were collected using questionnaires. The respondents were 22–88 years, with a median age of 60.25 (13.51). Seventy-nine (67.5%) respondents were male. Most respondents reported a low-to-moderate postoperative pain level (average pain score = 2.02; range: 0–10). The average anxiety score was 4.68 (standard deviation [SD] = 4.2), and the average depression score was 6.91 (SD = 4.52; range: 0–21). The average sleep efficiency index was 70.4% (SD = 10.74%). Most (95.7%) respondents had a sleep efficiency index below 85%, indicating that most patients did not sleep well in the ICU. Linear regression analysis showed that the key predictors of the sleep quality of open-heart patients in the ICU were wound pain (β = −1.9) and noise disturbance (β = −1.86). These results provide information on sleep quality and the factors affecting postoperative patients in the ICU. These findings can be used as a reference for developing relevant interventions.

## 1. Introduction

During open-heart surgery, patients receive a series of respiratory inhibitors, such as general anesthesia, sedatives, and postoperative analgesics, which can lead to the relaxation of the upper airway dilator muscles, reduce the ventilatory response to hypoxia and hypercapnia [1], and increase the likelihood of postoperative cardiovascular complications. Therefore, open-heart patients are generally admitted to the intensive care unit (ICU) after surgery to be monitored and receive postoperative treatment if necessary. Good sleep quality facilitates recovery. However, open-heart patients often experience sleep problems in postoperative intensive care [2]. Common sleep problems are obstructive sleep apnea (OSA; prevalence = 19.2–50.2%) and delirium (prevalence = 30–50%) [3,4]; Chan et al. (2019) found a strong correlation between sleep deprivation and myocardial infarction, atrial fibrillation, stroke, and death within 30 days of surgery. Patients receiving postoperative intensive care have fragmented sleep patterns and significantly reduced Stage 3 (deep) and Stage 4 (REM) sleep [5]. Studies have shown that over half of ICU patients are sleep-deprived, with a third of patients reporting that their sleep problems persisted after discharge [6,7]. Sleep deprivation negatively impacts tissue repair, cell-mediated immunity, cognitive functions, and mental health [8]. It also has been associated with prolonged hospital stays and increased morbidity and mortality [9].

Although ICUs with a comprehensive range of therapeutic instruments and staffed with highly specialized medical personnel can provide for the medical needs of open-heart patients, they may be the source of sleep disturbances. Knowing patients’ sleep experiences and the factors affecting sleep can inform the development of interventions to improve the patient’s sleep quality. However, few studies have been published on the sleep experiences of open-heart patients in postoperative intensive care. Based on the studies of other patient groups, the factors affecting ICU patients’ sleep can generally be sorted into two broad categories. The first category includes physiological and therapeutic factors, such as pain, ventilation, and medications (sedatives). The second category is the psychological and environmental factors, such as anxiety, stress, unfamiliar environment, sensory deprivation, sensory overload, noise, light, and care activities [5,10,11]. Given the lack of studies on this topic, this study aims to investigate the sleep quality of open-heart patients in postoperative intensive care and factors that may affect their sleep, including demographics, disease characteristics, comorbidity index, pain, anxiety, depression, and the ICU environment.

## 2. Materials and Methods

### 2.1. Design

This is a cross-sectional observational study. Data were collected using structured questionnaires. The hospital’s institutional review board (IRB) approved the study, where the participants were recruited.

### 2.2. Participants

A consecutive sample of postoperative patients was recruited from a general hospital’s cardiovascular intensive care unit (ICU) in Taiwan. All patients meeting the following eligibility criteria were recruited between June 2020 and March 2021. The inclusion criteria were patients (1) aged 20 years or older, (2) who underwent open-heart surgery, and (3) who were fluent in Mandarin or Taiwanese. The exclusion criteria were patients (1) having emergency surgery, (2) developing postoperative complications, (3) being diagnosed with severe mental illness, (4) being intubated or having tracheotomy, and (5) being unable to communicate verbally or through writing.

The required sample size was estimated using G Power v. 3.2. (Heinrich-Heine-Universität Düsseldorf, Düsseldorf, Germany). The F-test’s multiple linear regression with a fixed model and R2 increase was chosen, with the effect size (f2), significance level, and power being 0.20, 0.05, and 0.80, respectively. The number of predictor parameters was set to 15. Under these conditions, the estimated sample size required was 110.

### 2.3. Data Collection and Instruments

One of the researchers approached the potential participants at the bedside in the morning (between 11 am to noon) on the second or third day after open-heart surgery. After signing informed consent, the participants answered the study questionnaire independently but could ask the researcher to clarify anything they found unclear. For those with poor eyesight or who could not read or answer the questionnaire independently, the researcher read the questionnaire items aloud to them and wrote down their verbal responses. 

Participants took about 30 min to complete the study questionnaire. The 45-item self-report questionnaire included demographic questions (age, gender, education, marital status, and employment status), the pain numeric rating scale, the Chinese version of the hospital anxiety and depression scale (HADS), the Richards–Campbell sleep questionnaire (RCSQ), the sleep in the intensive care unit questionnaire (SICUQ), and one open-ended question. The following data were collected from each participant’s medical record: the types of surgery, history of intensive care, history of sleeping problems, use of sedatives or sleeping medication, and comorbidities. 

Comorbidities were further calculated as the Charlson Comorbidity Index (CCI). The CCI measures the comorbidity level by considering the number and severity of 19 predefined comorbid conditions, each of which is allocated a weighted value of 1, 2, 3, or 6, depending on the relative risk [12]. The sum of the weighted values represents the respondent’s CCI, with a higher score denoting a higher morbidity rate. Specifically, the morbidity rate increases by a factor of 1.4 per point. When merging the age and comorbidity into an age-adjusted CCI value, the estimated relative risk increased by 1.45 [13]. Katz et al. (1994) tested the self-reported CCI against the medical-record–based CCI with an intraclass correlation coefficient (ICC) of at least 0.40 and obtained test–retest reliability of 0.91 [14]. The CCI has also been valid and reliable in several mortality rate studies [15]. Quan et al. (2011) tested the CCI against the inpatient mortality rate of six countries and obtained a regional validity of 0.73–0.88 [16]. 

The pain numeric rating scale was used to measure the average wound pain experienced by the patients the night before [17]. On a linear scale ranging from 0 (no pain) to 10 (severe pain), the respondents were asked to select a value that best described the pain they experienced [18]. Values 1–4 denoted slight pain, 5–6 denoted moderate pain, and 7–10 denoted severe pain. The pain numeric rating scale has excellent discriminant validity (coefficient alphas: 0.76–0.91) and sensitivity (*t* = 5.88, *p* < 0.001) [19,20].

The Chinese version of HADS was used to assess the level of anxiety and depression in patients [21]. HADS includes 14 items divided into two subscales: anxiety and depression. Each subscale contains seven items with four options on a scale of 0–3. Therefore, the scores for each subscale range from 0 to 21, with higher scores indicating more significant anxiety or depression. A score of <8 represents no anxiety or depression (non-case), 8–10 represents suspected anxiety or depression (suspected case), and ≥11 represents confirmed anxiety or depression (confirmed case). The Chinese version of HADS showed favorable psychometric properties in patients with coronary heart disease. Cronbach’s alpha was 0.85, and the test–retest reliability was 0.90. It also showed good concurrent validity with the Short Form-36 Health Survey [22]. In the current study, Cronbach’s alpha for the anxiety and depression subscales were 0.84 and 0.74, respectively.

The Chinese version of the RCSQ measured the participants’ sleep quality. The RCSQ was explicitly developed for ICU patients [23]. It included five items: sleep depth, falling asleep, awakening, restorative sleep, and sleep quality. Each item was scored on a visual analog scale (VAS), where 0 meant poor sleep and 100 meant good sleep. The scale score was the average of all the items. A higher score indicated better sleep, 0–25 indicated abysmal sleep, and 76–100 indicated good sleep [23]. The RCSQ score can be converted to the sleep efficiency index (SEI) using the following equation: 46.88 + (0.39 * RCSQ). SEI represents the percentage of total sleep time during a specific period. An SEI of over 85% represents good sleep quality. The principal component factor analysis yielded an eigenvalue of 3.61 and a variance of 72.2% [24]. Internal consistency was achieved when Cronbach’s alpha coefficient was 0.90, which in this study was 0.93.

The SICUQ measured the participants’ sleep quality at home and in the ICU [25]. The scale had 20 items. Six items assessed the sleep disturbance by health care activities, including nursing interventions, diagnostic testing, vital signs, blood samples, and the administration of medications. Each item was rated on a numeric rating scale of 1 (no disruption) to 10 (significant disruption). The average score across the seven items represented the severity of sleep disturbances caused by healthcare activities. The higher the score, the more severe the interference. There were also 11 items to measure noise interference with sleep, including the heart monitor alarm ventilator alarm, ventilator, oxygen finger probe, talking, IV pump alarm, suctioning, nebulizer, doctor’s beepers, television, and telephone. Each item was rated on a scale of 1 (no disruption) to 10 (significant disruption). The average score of the 10 items represented the severity of noise disturbance. The higher the score, the more severe the disturbance [25]. One item measured “sleep quality at home” (1–10), and one item measured “sleep quality in the ICU” (1–10). Higher scores represented better sleep quality. One item measured “daytime sleepiness in the ICU” (1–10). Higher scores represented more severe daytime sleepiness. The scale demonstrated good psychometric properties in previous studies [26]. In this study, Cronbach’s alpha for the total scale, the healthcare activity subscale, and the noise disturbance subscale were 0.90, 0.86, and 0.86, respectively.

To explore the participants’ perceptions of the factors that affected their sleep quality, we also included an open-ended question: “please describe the most influential factor that affected your sleep last night?”

### 2.4. Statistical Analysis

Descriptive statistics were used to describe the central tendency and dispersion of the study variables. The relationships between sleep quality and other study variables were analyzed by an independent sample t-test, one-way analysis of variance (ANOVA), and Pearson correlation analysis. The multiple linear regression analysis was used to explore the critical predictors of sleep quality. Answers to the open-ended question were transcribed verbatim, and a content analysis was conducted to categorize the content of the answers.

## 3. Results

### 3.1. Demographics, Disease Characteristics, Pain, Anxiety, and Depression

Two hundred and fourteen patients were screened for eligibility; six did not meet the eligibility criteria and were excluded. Of these six patients, one underwent emergency surgery, one was intubated, one died after surgery, and three developed postoperative complications. The 118 potentially eligible patients were approached; one of them refused to participate. A total of 117 patients provided signed informed consent and participated in the study. 

The participants were 22–88 years, with a mean age of 60.25 (standard deviation [SD] = 13.51). Most of them were male (*n* = 79, 67.5%), married (*n* = 92, 78.6%), and employed (*n* = 60, 51.3%). Seventy-one (60.7%) participants had undergone valve repair or replacement surgery, and 46 (39.3%) had undergone coronary artery bypass graft (CABG) surgery. Sixty-eight (58.1%) participants received hypnotics in the ICU. Thirty-six (32.5%) participants had a history of sleep problems; of these, 15 (12.8%) had taken hypnotics at home. Twenty-nine patients (24.8%) had prior ICU experience. The average CCI was 0.63 (SD = 0.93). The average wound pain during the previous evening was 2.9 (SD = 2.0). The average anxiety score was 4.7 (SD = 4.2), and the average depression score was 6.9 (SD = 4.5; Table 1).

### 3.2. Sleep Quality

The participants’ average sleep quality score was 59.9 (SD = 23.5) on the RCSQ. This converted into an SEI of 70.2% (SD = 9.2). An SEI of over 85% represented good sleep. Only five (4.3%) of the respondents had an SEI over 85%, suggesting that most patients slept poorly in the ICU (Table 2). On a scale of 1–10, the participants’ average sleep quality was 7.2 (SD = 2.5) at home and 6.0 (SD = 2.6) in the ICU. The average daytime sleepiness score was 5.4 (SD = 2.4). These suggest that the patient’s sleep quality in the ICU was poorer than at home, and they tended to be moderately sleepy during the day. The average score for healthcare activity disturbance was 2.0 (SD = 1.5). The average score for noise disturbance was 1.9 (SD = 1.2; Table 1).

### 3.3. Factors Affecting Sleep Quality

The results of the T-tests and one-way ANOVAs showed no significant difference in sleep quality among the participants with different demographics and disease characteristics (Table 2). We then performed a Pearson’s correlation analysis to analyze the associations between the RCSQ scores and pain, anxiety, and depression. The results of the person correlation analysis showed that patients’ sleep quality was significantly and negatively correlated with depression (r = −0.26, *p* = 0.004) and noise disturbance (r = −0.32, *p* < 0.001). Patients who experienced less depression and noise disturbance thus had better sleep quality (Table 3).

All the study variables were entered into a stepwise linear regression model as the independent variables for predicting the participants’ sleep quality. The results showed that noise disturbance (B = −5.8) and depression (B = −1.14) were significant predictors of the participants’ sleep quality in ICU (Table 4).

### 3.4. Participant-Perceived Factors Affecting Sleep

Eighty-seven participants answered the open-ended question about the factors that affected their sleep. Their responses were transcribed verbatim. These contents were analyzed to categorize the main factors perceived by the participants as affecting their sleep. We identified the critical self-reported factors by tallying the respondents’ answers. The most commonly mentioned factors were wound pain (*n* =17), chest tube or aspirator noise (*n* = 10), psychological factors, such as anxiety, nervousness, restlessness, concern about postoperative care, and unfamiliar environment (*n* = 9), the noise made by staff (*n* = 8), environmental noise (*n* = 8), position-related factors such as soreness, inability to turn over, and too many tubes (*n* = 7), the noise from other patients (*n* = 6), drainage tube (*n* = 5), personal physiological factors (*n* = 5), such as the need to urinate, sputum, hiccups, sleepiness, and lethargy, and other tube-related factors (*n* = 4) such as a urinary catheter, sore throat (*n* = 3), and persistent insomnia (*n* = 3).

## 4. Discussion

The study results show that open-heart patients experienced moderate sleep disturbances in postoperative intensive care. Our participants’ average RCSQ score was 59.9, which was consistent with a score of 50.11–61.29 in open-heart patients reported postoperative on days 1 to 4 by Yayla and Özer [27]. The average sleep quality score on the SICUQ was 7.2 at home and 6.0 in the ICU on a scale of 0 to 10. These scores were similar to those reported by Cicek et al. [28], who surveyed 100 patients in a coronary ICU and recorded an average score of 7.4 at home and 5.4 in the ICU. These findings suggest that patients’ sleep quality tends to be lower in the ICU than at home. Sleep quality at night strongly affects mental state during the day. Poor sleep has resulted in increased daytime fatigue or sleepiness [27]. In this study, the patients’ average daytime sleepiness score was 5.4 (SD = 2.4), suggesting moderate diurnal sleepiness.

The factors affecting sleep quality can be discussed from an internal (or patient-specific) versus an external perspective. Internal factors may include psychological state and pain. External factors may include disturbance from healthcare activities and noise. The patient’s psychological state, including anxiety and depression, may affect his/her sleep quality. Our results showed that depression was negatively associated with sleep quality, supporting the fact that depression affected sleep quality. While there was also a negative correlation between anxiety and sleep quality (r = −0.21, *p* = 0.24), this relationship did not reach statistical significance (alpha = 0.05/8) when a Bonferroni correction was used to counteract the problem of multiple comparisons. In the open-ended question, however, patients reported that anxiety, nervousness, restlessness, concern about postoperative care, and the unfamiliar environment affected their sleep quality. In this study, there were 17 (14.5%) suspected and 11 (9.4%) confirmed cases of anxiety, 23 (19.7%) suspected, and 27 (23.1%) confirmed cases of depression. Psychological factors, therefore, should not be overlooked. Having familiar caregivers provide reassurance, psychological support, and clear explanations can help alleviate doubt and concerns, thus improving the sleep quality of open-heart patients in postoperative intensive care.

Postoperative pain and healthcare activities in the ICU may negatively affect sleep quality in patients with open-heart surgeries; conversely, sleep deprivation increases pain sensitivity [29]. Our quantitative data do not support that pain and healthcare activities affected sleep quality because their associations were not statistically significant. However, from the open-ended question, we found that the participants perceived wound pain, post-sternotomy supine position, chest tubes, drainage tubes, and noise from medical equipment as the primary sources of sleep disturbance. Despite the treatment of pain medications, the participants’ average postoperative wound pain score ranged from 2.9 to 4.4, indicating an area of concern. Therefore, postoperative wound stabilization and pain control should be considered while improving sleep quality in open-heart patients. The chest and other drainage tubes should be removed as soon as possible. 

The noise emitted by ICU equipment also affected the patients’ sleep quality. A noise disturbance was negatively correlated with sleep quality and predicted sleep quality. These results support those of Simons et al. (2018), who found that noise volume negatively correlated with ICU patients’ self-reported sleep quality [30]. Further, Czempik et al. (2020) found that ICU noise volume was negatively correlated with patients’ sleep time and quality, with 17% of respondents reporting that environmental noise was the main factor affecting their sleep quality [31]. Chaudhary et al. (2020) found that 76.6% of ICU patients (*n* = 60) experienced poor sleep and that pain (33.3%), noise (31.7%), and light (3.3%) were the most critical factors [32]. Among the noise-related factors reported in the present study, a notable factor was “noise from other patients”. ICU patients must be placed in a designated location and be constantly monitored. Although partitions separate them, several noise sources may affect their sleep quality, including neighboring patients experiencing delirium or changes in condition, displaying warnings, staff conversations, and environmental noise. Reducing noise levels may effectively alleviate environmental noise, improving the sleep quality of open-heart patients in postoperative intensive care [32]. These include reducing conversation volume, lowering the volume of machine warning tones, using white noise machines, or providing earplugs.

Sleep quality was investigated using two self-reported questionnaires, which cannot objectively measure sleep parameters. There may also be a particular memory bias in the questionnaire data. Nonetheless, these limitations do not affect our findings that open-heart patients experience moderate sleep disturbances during postoperative intensive care. In addition, we presented them with an open-ended question to allow the respondents to self-report factors that affected their sleep quality. Their answers revealed many factors that were not in the questionnaire. A semi-structured questionnaire survey would allow researchers to obtain a comprehensive dataset to elucidate the sources and the severity of sleep disturbances. Therefore, a research approach that combines objective and subjective assessment tools should be used to examine sleep conditions. Another limitation of this study was that we did not examine the effects of medication on sleep quality. Many types of medication hinder sleep, and sedatives and analgesics are commonly administered in the ICU. Some may even affect normal sleep physiology and the content of sleep. Therefore, medication is a critical factor that should be considered in future research. The final limitation was that the participants were recruited from the cardiovascular surgery ICU of one hospital. Our findings may not be generalizable to open-heart patients in other ICUs. We recommend that this research be repeated with patients from other hospitals to ensure generalizability. 

## 5. Conclusions

The impact of poor sleep quality on open-heart patients is multifaceted, and many factors affect the sleep quality of patients in the ICU. Based on a quantitative assessment and an open question, we found that environmental noise and depression were the key factors that affected the sleep quality of open-heart patients in postoperative intensive care. The most important of these, both objectively and subjectively, was wound pain. To improve the sleep quality of open-heart patients, physicians should assess the sleep conditions of their patients and incorporate pain assessment and control into their patients’ sleep plans. In addition, formulating strict nighttime standard operating procedures that reduce noise and light, minimize unnecessary disturbances, and remove unnecessary tubes will significantly improve the patient’s sleep quality. Intelligent monitoring systems and remote monitoring applications should reduce instrument noise. Strategies to increase daytime activity levels may help to reduce daytime sleepiness and improve sleep quality by maintaining circadian rhythms. Nurses should be trained to assess and detect emotional distress and provide support, and steps should be taken to improve the patient’s physical and psychological comfort. All of this would help to improve sleep quality in the ICU.

Although many interventions may improve the patient’s sleep quality, sleep factors differ according to the ailment and situation. Therefore, each situation should be analyzed independently to provide specific care recommendations. Although the factors highlighted in this study were not strong predictors of sleep quality in these open-heart patients in postoperative intensive care, and many other potential factors were not discussed here, the topic is critical. In the future, researchers should consider using objective tools and performing qualitative interviews to reach more comprehensive conclusions about the sleep experiences of open-heart patients and improve their sleep quality.

## Figures and Tables

**Table 1 healthcare-10-02311-t001:** Demographic characteristics, disease characteristics, pain, anxiety, depression, and sleep quality of the study participants (N = 117).

Variable		N	%	Range
Age (mean; SD)		60.25	13.51	22–88
Sex	Female	38	32.5	
Education	Male	79	67.5	
	Elementary or lower	24	20.5	
	Middle school	19	16.2	
	High/vocational school	26	22.2	
	College/university	36	30.8	
	Graduate school	12	10.3	
Marital status	Single	12	10.3	
	Married	92	78.6	
	Divorced/widowed	13	11.1	
Employment status	Full-/part-time	60	51.3	
	Unemployed	17	14.5	
	Retired	40	34.2	
Type of surgery	Valve repair	71	60.7	
	CABG	46	39.3	
Hypnotic drugs	No	47	40.2	
	Yes	68	58.1	
Hx of sleep problems	No	79	67.5	
	Yes	38	32.5	
Prior ICU experience	Yes	88	75.2	
	No	29	24.8	
Charlson Comorbidity Index		0.63	0.93	0–6
Wound pain		2.9	2.0	0–10
Anxiety		4.7	4.2	0–17
Depression		6.9	4.5	0–20
Sleep quality at home		7.2	2.5	1–10
Care activity disturbance		2.0	1.5	1–7.7
Noise disturbance		1.9	1.2	1–6.6
Sleep quality		59.9	23.5	9.5–100
SEI		70.2	9.2	50.6–85.9

ICU—intensive care unit; SD—standard deviation; CABG—coronary artery bypass graft; SEI—sleep efficiency index.

**Table 2 healthcare-10-02311-t002:** The difference in sleep quality among participants with different demographics and **disease** characteristics (N = 117).

Variable		N	Sleep Quality		
			Mean	*t*/F	*p*
Sex	Female	38	62.31	0.78	0.44
	Male	79	58.71		
Education	Lower school	24	61.00	0.79	0.54
	Middle school	19	57.36		
	High school	26	64.33		
	College	36	60.41		
	Graduate school	12	50.38		
Marital status	Single	12	53.17	0.58	0.56
	Married	92	60.39		
	Divorced/widowed	13	62.45		
Employment status	Full-/part-time	60	58.25	0.43	0.65
	Unemployed	17	59.06		
	Retired	40	62.68		
Type of surgery	CABG	46	60.3	0.16	0.87
	Valve repair	71	59.6		
Hypnotic drugs	No	47	61.00	0.34	0.74
	Yes	68	59.48		
Hx of sleep problems	No	79	60.22	0.23	0.82
	Yes	38	59.16		
Prior ICU experience	No	88	58.78	−0.88	0.38
	Yes	29	63.22		

*t*—the value of independent test; F—the value of one-way analysis of variance.

**Table 3 healthcare-10-02311-t003:** Correlation coefficients between sleep quality and other study variables (N = 117).

	Age	CCI	Wound Pain	Anxiety	Depression	Care Activity	Noise Disturbance	Sleep at Home	Sleep Quality
Age	1.00								
CCI	0.17	1.00							
Wound pain	−0.20 *	−0.12	1.00						
Anxiety	0.18	0.05	0.28 *	1.00					
Depression	−0.01	−0.01	0.29 *	0.63 *	1.00				
Care activity	−0.08	0.04	0.16	0.27 *	0.18	1.00			
Noise disturbance	−0.11	0.03	0.08	0.24	0.15	0.60 *	1.00		
Sleep at home	−0.15	−0.15	0.08	−0.22	−0.14	−0.21	−0.07	1.00	
Sleep quality	0.04	0.08	−0.13	−0.22	−0.26 *	−0.17	−0.32 *	0.11	1.00

* Correlation is significant at the 0.006 level (2-tailed) based on the Bonferroni correction to counteract the multiple comparisons problem (alpha = 0.05/8 = 0.006).

**Table 4 healthcare-10-02311-t004:** Predictors of sleep quality (N = 117).

Variable	B	*t*	*p*	95% CI	VIF	Adjusted R^2^	F
Final model						0.14	10.2
Constant	78.7	16.9	<0.001	[69.5, 88.0]			
Noise disturbance	−5.8	−3.3	0.001	[−9.3, −2.3]	1.02		
Depression	−1.1	−2.5	0.013	[−2.0, −0.2]	1.02		

Note: All the study variables were entered into a stepwise linear regression model. Criteria: probability-of-F-to-enter ≤ 0.050. Abbreviations: VIF—the variance inflation factor.

## Data Availability

Data will be available from the corresponding author upon reasonable request.

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
