# Peer review of "Factors Influencing Sleep Quality in Open-Heart Patients in the Postoperative Intensive Care Unit"

_healthcare, 2022, doi:10.3390/healthcare10112311_

Round 1
Reviewer 1 Report
Dear Author
Your manuscript entitled "Factors Influencing Sleep Quality in Open-heart Patients in the Postoperative Intensive Care Unit" was well-presented. As you mentioned in the limitation of the study, the effects of medication on sleep quality should be determined to give an objective evidence.
I have some questions and comments:
- You used many questionnaires as a study tool. In total, how many questions were there? How long the patient took to answer? What time you approached? Mention in detail in the methodology. Will filling the questionnaire itself not affect the sleep quality of patients?
- The open-ended question: "please describe the most influential factor in your sleep last night?" This question means what influences the patient to sleep. But you mean what affected their sleep? Did you explain the exact meaning to the patients? The question could have been "please describe the most influential factor that affected your sleep last night?"
- In the discussion lines 224-226 should be removed.
- In the conclusion you mentioned that strict nighttime standard operating procedures should be followed to improve sleep quality. So you measured the sleep quality at night only? What about the day time sleep?
Author Response
Thank you for thoroughly reviewing our manuscript. Your recommendations have been extremely helpful. We have updated the text in the manuscript per your recommendations and indicated the changes using red-colored text. Please find below a point-by-point response to your comments. Each critique is followed by ‘Response’ indicating our action. We believe that we have addressed all the critiques from reviewers. The entire manuscript has been checked for sentence structure and grammatical errors. If further revision is needed, please let us know and we shall promptly. Thanks again for considering our manuscript entitled Factors Influencing Sleep Quality in Open-heart Patients in the Postoperative Intensive Care Unit.
Sincerely,
Tsae-Jyy Wang
National Taipei University of Nursing and Health Sciences
Reviewer 1
- You used many questionnaires as a study tool. In total, how many questions were there? How long the patient took to answer? What time did you approach? Mention in detail in the methodology. Will filling out the questionnaire itself not affect the sleep quality of patients?
Response: The study questionnaires consist of a total of 45 self-report items and take less than 30 minutes to complete. Patients completed the questionnaires in the morning (between 11 am to noon) on the second or third day after open-heart surgery. Completing the questionnaires itself should not affect the patient’s sleep quality (lines 85-93).
- The open-ended question: "please describe the most influential factor in your sleep last night?" This question means what influence the patient to sleep. But you mean what affected their sleep? Did you explain the exact meaning to the patients? The question could have been "please describe the most influential factor that affected your sleep last night?"
Response: This confusion is due to the English translation problem. A more accurate translation should be "please describe the most influential factor that affected your sleep last night?"
- In the discussion lines 224-226 should be removed.
Response: In the discussion, lines 224-226 have been removed
- In the conclusion, you mentioned that strict nighttime standard operating procedures should be followed to improve sleep quality. So you measured the sleep quality at night only? What about daytime sleep?
Response: To address this issue, the following sentence has been added in the conclusion section “Strategies to increase daytime activity levels may help to reduce daytime sleepiness and improve sleep quality by maintaining circadian rhythms.” (lines 320-322)
Reviewer 2 Report
This is an important study and I congratulate authors on their great work in putting this research together.
1. In the abstract, the sentence ("Only five (4.3%) respondents had a sleep index exceeding 85%, indicating that the 21 sleep quality of these patients in postoperative intensive care was poor.") can be misinterpreted to indicate that only 4.3% had sleep problems -- perhaps rewording this sentence will be helpful. By "these patients", authors mean patients in the study, but it can be misinterpreted as these 4.3%.
2. Is there information about the reliability of wound pain and noise disturbance measures (subscales)? Can readers see the measures (e.g., items)?
3. It seems like there were multiple tests/investigations to find significant predictors of sleep quality. This requires multiple comparison adjustments (e.g., Bonferroni, or using adjustments based on False Discovery Rates).
4. I wonder if factors (influencing sleep quality) can be discussed in terms of internal (or patient-specific) vs. external. I wasn't sure what noise disturbance represents exactly (clarification would be great), but it sounds like an external factor -- more like an environmental or macro-level effect.
Author Response
Dear reviewer,
Thank you for thoroughly reviewing our manuscript. Your recommendations have been extremely helpful. We have updated the text in the manuscript per your recommendations and indicated the changes using red-colored text. Please find below a point-by-point response to your comments. Each critique is followed by ‘Response’ indicating our action. We believe that we have addressed all the critiques from reviewers. The entire manuscript has been checked for sentence structure and grammatical errors. If further revision is needed, please let us know and we shall promptly. Thanks again for considering our manuscript entitled Factors Influencing Sleep Quality in Open-heart Patients in the Postoperative Intensive Care Unit.
Sincerely,
Tsae-Jyy Wang
National Taipei University of Nursing and Health Sciences
- In the abstract, the sentence ("Only five (4.3%) respondents had a sleep index exceeding 85%, indicating that the 21 sleep quality of these patients in postoperative intensive care was poor.") can be misinterpreted to indicate that only 4.3% had sleep problems -- perhaps rewording this sentence will be helpful. By "these patients", authors mean patients in the study, but it can be misinterpreted as this 4.3%.
Response: The sentence has been revised: "The majority (95.7%) of respondents had a sleep efficiency index below 85%, indicating that most patients did not sleep well in the ICU.”
- Is there information about the reliability of wound pain and noise disturbance measures (subscales)? Can readers see the measures (e.g., items)?
Response: The wound pain was measured with a pain numeric rating scale. Its validity and sensitivity were described in lines 115-116. The SICUQ can be seen in the [25] reference. However, the information regarding the items has also been added in the manuscript to clarify the questionnaire. “Six items assessed sleep disturbance by health care activities, including nursing interventions, diagnostic testing, vital signs, blood samples, and administration of medications. ….There are also 11 items to measure noise interference with sleep, including heart monitor alarm, ventilator alarm, ventilator, oxygen finger probe, talking, IV pump alarm, suctioning, nebulizer, doctor’s beepers, television, and telephone.” (lines 140-148) The reliability of the total scale, the healthcare activity subscale, and the noise disturbance subscale have been added to the manuscript. (lines 154-155)
- There seem to be multiple tests/investigations to find significant predictors of sleep quality. This requires multiple comparison adjustments (e.g., Bonferroni, or using adjustments based on False Discovery Rates).
Response: The Bonferroni correction was used to counteract the multiple comparisons problem in Table 3. (lines 154-155)
- I wonder if factors (influencing sleep quality) can be discussed in terms of internal (or patient-specific) vs. external. I wasn't sure what noise disturbance represents exactly (clarification would be great), but it sounds like an external factor -- more like an environmental or macro-level effect.
Response: Factors affecting sleep quality can be discussed from an internal (or patient-specific) versus external perspective. Internal factors may include pain and psychological status. External factors may include disturbance from healthcare activities and noise. This information has been added to the discussion section on lines 247-249.